# Association between *PYTPN22* rs2476601, *VEGF* rs833070, *TNFAIP3* rs6920220 Polymorphisms and Risk for Rheumatoid Arthritis in Early Undifferentiated Arthritis Patients: A Pilot Study

**DOI:** 10.3390/medicina59101824

**Published:** 2023-10-13

**Authors:** Regina Sakalyte, Sigita Stropuviene, Gabija Jasionyte, Loreta Bagdonaite, Algirdas Venalis

**Affiliations:** 1The Clinic of Rheumatology, Traumatology Orthopaedics and Reconstructive Surgery, Institute of Clinical Medicine of the Faculty of Vilnius University, M. K. Čiurlionio Str. 21, 03101 Vilnius, Lithuania; sigita.stropuviene@santa.lt (S.S.); gabija.jasionyte@santa.lt (G.J.); algirdas.venalis@imcentras.lt (A.V.); 2State Research Institute Centre for Innovative Medicine, Santariškių g. 5, 08406 Vilnius, Lithuania; 3Department of Physiology, Biochemistry, Microbiology and Laboratory Medicine, Faculty of Medicine, Vilnius University, M. K. Čiurlionio Str. 21, 03101 Vilnius, Lithuania; loreta.bagdonaite@santa.lt

**Keywords:** rheumatoid arthritis, early undifferentiated arthritis, single-nucleotide polymorphisms

## Abstract

*Background and Objectives*: About 40% of early undifferentiated arthritis (UA) progresses to rheumatoid (RA) or other chronic arthritis. Novel diagnostic tools predicting the risk for this progression are needed to identify the patients who would benefit from early aggressive treatment. Evidence on the role of single-nucleotide polymorphisms (SNPs) in the development of RA has emerged. The aim of our study was to investigate the association between rs2476601, rs833070, and rs6920220 SNPs and UA progression to RA. *Materials and Methods*: Ninety-two UA patients were observed for 12 months. At study entry, demographic and clinical characteristics were recorded, musculoskeletal ultrasonography was performed, and blood samples were drawn to investigate levels of inflammatory markers, rheumatoid factor (RF), anti-citrullinated protein antibodies (anti-CCP)detect SNPs. After 12 months, UA outcomes were assessed, and patients were divided into two (RA and non-RA) groups. The association between the risk of progression to chronic inflammatory arthritis and analyzed SNPs was measured by computing odds ratios (OR). *Results*: After a 12-month follow-up, 27 (29.3%) patients developed RA, and 65 (70.7%) patients were assigned to the non-RA group. The arthritis of 21 patients (22.8%) from the non-RA group resolved completely, while the other 44 (47.2%) patients were diagnosed with another rheumatic inflammatory disease. The patients who developed RA had a significantly greater number of tender and swollen joints (*p* = 0.010 and *p* = 0.021 respectively) and were more frequently RF or anti-CCP (*p* < 0.001), and both RF and anti-CCP positive (*p* < 0.001) at the baseline as compared with the patients in the non-RA group. No significant association between rs2476601 (OR = 0.99, *p* = 0.98), rs833070 (OR = 1.0, *p* = 0.97), and rs6920220 (OR = 0.48, *p* = 0.13) polymorphisms and the risk of developing RA were found. *Conclusions*: No association between analyzed SNPs and a greater risk to progress from UA to RA was confirmed, although patients with rs6920220 *AA* + *AG* genotypes had fewer tender joints at the disease onset.

## 1. Introduction

Early undifferentiated arthritis (UA) is an autoimmune inflammatory joint disease; it is a very common diagnosis among individuals assessed in rheumatology practice [1], and it displays signs and symptoms of inflammatory arthritis that do not meet the classification criteria for any specific rheumatic disease [2,3]. UA can be an early manifestation of defined arthritis such as rheumatoid arthritis (RA), psoriatic arthritis (PsA), or ankylosing spondylitis (AS), or it can be a self-limited syndrome of an unknown cause that resolves on its own [4]. While up to 60% of UA patients can remit spontaneously, the remaining ones progress to RA or other chronic arthritis, often within one year [2,5]. There is strong evidence that early aggressive treatment of UA patients can postpone progression to RA, preventing further joint damage and thus preserving functional ability [6]; drug-free remission can also be achieved [7]. Therefore, it is crucial to develop new diagnostic tools to predict which patients with UA are likely to develop chronic erosive arthritis such as rheumatoid arthritis so that early aggressive treatment can be specifically targeted at this patient group.

The current knowledge suggests the presence of specific autoantibodies (rheumatoid factor (RF)anti-citrullinated, anti-carbamylated, and anti-acetylated peptide antibodies) [8], elevated erythrocyte sedimentation rate (ESR), C- reactive protein values [9,10], and tenosynovitis [8] as possible predictors of progression to RA. Although genetic factors are thought to be responsible for up to 66% of predisposition to RA [11,12], and a large number of genetic variants have been associated with RA [13], there are only few studies that have examined genetic variants as predictors of progression from UA to RA [13,14]. A human genome-wide association analysis has identified a large number of SNPs that are associated with RA [15]. A meta-analysis of over 100,000 individuals of European and Asian descent (29,880 RA cases and 73,758 controls) evaluated ~10 million SNPs and identified 101 polymorphisms associated with RA risk. In addition, a few SNPs (rs2476601, rs833070, and rs6920220) were found to be associated with RA diagnostic markers [16,17,18]. 

rs2476601 is polymorphism in the *non-receptor type 22* gene (*PTPN22*), which encodes lymphoid-specific phosphatase (OR 1.06) [19]. This protein is known to be important in the etiopathogenesis of RA and other autoimmune rheumatic diseases [20,21]. A meta-analysis confirmed that RA patients who had elevated anti-CCP and/or RF value carried *T* allele and *CT + TT* genotypes significantly more frequently than healthy controls did [16]. 

rs833070 is located in the *VEGF* gene, which is known to be involved in autoimmune disease etiopathogenesis [22]. The analysis showed that RA patients who had the *AA* genotype had higher VEGF levels and DAS 28 values as compared with those who had *AG* or *GG* genotypes. On the other hand, in patients who had *AG* or *GG* genotypes, a US analysis revealed higher synovitis and more active blood circulation as compared with the patients who had the *AA* genotype [23]. There are a few more studies that detected the association between this polymorphism and VEGF levels in sera of RA patients [18]. 

Polymorphism rs6920220 is in the *tumor necrosis factor-alpha inducible protein 3* (*TNFAIP3*) gene [24]. Meta-analysis showed the association between this SNP and RA [24]; another study confirmed this association particularly with positive anti-CCP and RF profile [17].

The hypothesis of this undifferentiated arthritis cohort pilot study is as follows: rs2476601, rs833070, and rs6920220 SNPs can be prognostic markers for UA progression to RA.

The aim of this study was to investigate the rs2476601, rs833070, and rs6920220 SNPs’ relationship with clinical and laboratory parameters that are applied every day in early undifferentiated arthritis differential diagnoses and to determine if tested SNPs are associated with UA progression to RA.

## 2. Patients and Methods

### 2.1. Patients

Ninety-two UA patients were recruited to participate in the prospective UA cohort study at Vilnius University Hospital, Santaros Klinikos, Rheumatology Center. The Vilnius Regional Biomedical Research Ethics Committee has approved this study (permit No. 158200-16-859-368). All patients signed informed consent before they were enrolled in the study. Patients were enrolled if they were at least 18 years of age, had no established inflammatory arthritis diagnosis at study entry [25,26,27,28,29], and had at least one swollen joint and the patient-reported duration of joint swelling was shorter than 12 months. Patients whose joint swelling was due to infectious arthritis, arthritis due to trauma, microcrystal, or paraneoplastic arthritis or osteoarthritis were excluded from the study. Participants were observed prospectively for 12 months in order to measure the outcomes of their UA. The disease outcome was the following: either the patient was diagnosed with inflammatory rheumatic disease based on rheumatic diseases classification criteria (American College of Rheumatology/European League Against Rheumatism (ACR/EULAR) 2010 rheumatoid arthritis classification criteria for RA [25], classification criteria for psoriatic arthritis (ClASsification for Psoriatic Arthritis (CASPAR)) [26], Assessment of Spondylarthritis International Society (ASAS) classification criteria for axial and peripheral spondylarthritis [27,28], 2012 Systemic Lupus International Collaborating Clinics criteria (SLICC’12) for systemic lupus erythematosus (SLE) [29], or the patient’s arthritis resolved (remission observed after 6-month follow-up (no swollen joints) without any need for steroids or DMARDs). Patients whose arthritis resolved completely had very low probability of progressing into chronic arthritis in the future. The final diagnosis of patients who did not attend a follow-up visit after 12 months was verified by reviewing medical electronic records. For statistical analysis based on the established diagnosis, first, all patients were divided into two groups: patients who developed RA (RA group) and those who did not (non-RA group). Second, patients in the non-RA group were subdivided into groups—patients whose arthritis resolved completely during the follow-up period and patients who developed other inflammatory rheumatic disease (e.g., psoriatic arthritis (PsA), ankylosing spondylitis (AS), SLE, etc.). 

### 2.2. Data and Sample Collection

At study entry, the patient‘s sex, age, height, weight, ethnicity, level of education, and smoking history were recorded. Body mass index (BMI) was calculated. The following clinical characteristics of the disease were recorded: comorbidities, patient-reported duration arthralgia and joint swelling in months, the duration of morning stiffness in minutes. A rheumatologist assessed 68 tender and 66 swollen joint counts (68 TJC/66 SJC) [30]; collected data on the patient’s and physician’s global assessment of disease activity and the patient’s pain assessment on 100 mm visual analogue scales (patient’s global VAS, physician’s global VAS, patient’s pain VAS); calculated the disease activity score 28 (DAS 28) based on the assessment of 28 tender and swollen joints; and recorded the patient’s global assessment, VAS, and erythrocyte sedimentation rate (ESR). The patients’ movement function or functional disability was evaluated via completion of the Health Assessment Questionnaire (HAQ) [31]. Ultrasound assessment of tender and swollen joints was also carried out. Synovitis, erosions, and power Doppler (PD) findings were scored using a scale of 0–3, and scores from each joint were summed up to calculate synovitis, power Doppler and erosion scores [32,33]. Blood samples drawn at baseline were used to determine ESR (Westergren method; reference value ≤ 20 mm/h) and C reactive protein (CRP) (turbidimetric method; reference value ≤ 5 mg/L) levels, as in routine rheumatologist clinical practice, to determine arthritis activity, as well as to determine vascular endothelial growth factor (VEGF) (enzyme-linked immunosorbent assay (ELISA) method, *IBL International*, Germany; reference value ≤ 43 pg/mL) levels and to extract DNA (for SNP detection: rs2476601, rs833070, and rs6920220). Anti-citrullinated protein antibodies (anti-CCP) (ELISA method, *Phadia AB*, Uppsala, Sweden; reference value < 5 U/mL), the rheumatoid factor (RF) (turbidimetric method; reference value < 30 kU/L), human leukocyte antigen B27 (HLA-B27) (flow cytometry method), anti-nuclear antibodies (ANA) (indirect immunofluorescence method; reference value < 1:40 titer) were all performed at the discretion of the treating rheumatologist as clinically indicated.

### 2.3. Sample Preparation and SNP Genotyping

Blood samples were collected in EDTA tubes and frozen. Genomic DNA was extracted from frozen whole blood using the Gentra Puregene Blood Kit (Qiagen, Hilden, Germany). Each patient sample was genotyped on the Illumina BeadLab1000 platform using the ImmunoChip V2 and the Infinium HD assay. Genotypes were identified using a score threshold of 0.15 in the Illumina BeadStudio software (https://www.illumina.com/Documents/products/datasheets/datasheet_beadstudio.pdf, accessed on 9 October 2023). Data cleaning was performed using the PLINK software program.

### 2.4. Statistical Analysis

Mean and standard deviation were used to describe the quantitative characteristics of research. Frequencies (n) and percentages (%) were used for qualitative characteristics. Data distribution normality was assessed using the Shapiro–Wilk normality test. Depending on applicable assumptions, Student’s *t*-test for independent samples was used to compare means of a particular qualitative characteristic of different samples. Differences in the qualitative characteristics of experimental groups were assessed using the Chi-square test. To compare nonparametric data sample means the Mann–Whitney U and the Kruskal–Wallis tests were performed. Correlations between all cohort data were calculated using Spearman’s correlation test. The association strength between the risk for UA to progress to chronic inflammatory arthritis and analyzed SNPs (rs2476601, rs833070, and rs6920220) were assessed by computing odds ratio (OR) with 95% confidence intervals (95%CI). All SNPs were tested for Hardy–Weinberg equilibrium, and all had minor allele frequencies (MAF) ≥10%. Statistical analysis and visualization were performed using Microsoft Office (Microsoft Corporation, Redmond, W.A., USA), SPSS (Version 26.0 IBM Corp, Armonk, NY, USA), and PLINK (Version 1.9). The selected level of significance was *p* < 0.05.

## 3. Results

### 3.1. Patient Characteristics

A total of 92 patients, mean age 44.25 ± 15.82 years, were enrolled in the present study; 61 (66.3%) were females; 30 (32.6%) reported a family history of rheumatic diseases. The mean UA duration was 5.27 ± 3.22 months. All of the enrolled patients (100%) were Caucasians. The mean numbers of swollen joints in 66/28 SJC were 3.17 ± 2.14 and 2.73 ± 1.60, respectively, and of tender joints in 68/28 TJC, 7.33 ± 5.77 and 5.37 ± 3.75, respectively. Median values of inflammation markers were as follows: ESR 32.55 [2–144] mm/h, CRP 17, 65 [0.16–144.30] mg/L; the median VEGF value was 516.16 [25.75–3438.25] pg/mL. Of 82 tested participants, 30 (36.1%) were positive for anti-CCP. At enrollment, 80 (86.96) of UA patients were DMARD-naïve. After a 12-month follow-up period had passed, as many as 27 (29.3%) patients had developed RA (RA group), and 65 (70.7%) patients were assigned to the non-RA group. Arthritis of 21 (22.8%) patients in the non-RA group resolved completely, and 44 (47.2%) of them were diagnosed with other rheumatic inflammatory diseases (PsA, AS, SLE, etc.) (Table 1).

The sociodemographic, clinical, and laboratory characteristics at the baseline and the comparison of data between UA patients who developed RA after 12 months and the non-RA group (whose arthritis resolved or who were diagnosed with other inflammatory rheumatic diseases) after a 12-month follow-up are presented in Table 1. Patients who developed RA within 12 months, as compared with those in the non-RA group, at the study baseline were present with a significantly greater number of tender (68/28 TJC) (*p* = 0.010/*p* = 0.021) and swollen (66/28 SJC) joints (*p* < 0.001/*p* < 0.001) and higher DAS 28 (ESR) scores (*p* < 0.005). The RA group vs. the non-RA one demonstrated significantly more frequent RF, or anti-CCP (*p* < 0.001), and or both RF and anti-CCP positive (*p* < 0.001). On the other hand, in the non-RA group, HLA-B27 was significantly more often expressed (*p* = 0.01). VEGF analysis revealed that patients who developed RA had higher VEGF levels as compared with those in the non-RA group, although it was not significant (*p* = 0.083) (Table 1). 

UA patients whose arthritis resolved completely as compared with the patients in the RA group at the baseline visit were more often diagnosed with active infection (*p* = 0.014). The number of swollen joints (66/28 SJC), and RF and anti-CCP positivity rates were significantly higher in the RA group as compared with the group of patients whose arthritis resolved (*p* < 0.001) (Table 1).

Compared with the group of patients who were later diagnosed with other inflammatory rheumatic (PsA, AS, etc.) diseases, the RA group patients had significantly higher numbers of swollen joints (66/28 SJC) and higher DAS 28 scores (*p* < 0.001) at the study baseline. The analysis of laboratory tests revealed that patients who developed RA had higher CRP (*p* = 0.036), RF, and anti-CCP values; additionally, RF and anti-CCP positive values (*p* < 0.001) and HLA-B27 were less expressed (*p* = 0.010) (Table 1).

Patients who developed RA had a higher rate of history of rheumatic diseases in blood relatives (as compared with the patients in the non-RA group and patients whose arthritis resolved); however, it was not statistically significant (*p* = 0.119 and *p* = 0.435 respectively) (Table 1).

The ultrasound analysis of tender and swollen joints of UA patients revealed that rates of synovitis, measured via power Doppler, at the onset of disease were significantly higher in patients who later developed RA as compared with those in the non-RA group, patients whose arthritis resolved, and patients who were diagnosed with another inflammatory rheumatic disease within a 12-month follow-up period (*p* < 0.001). Results were similar in the grade of erosions seen on US: RA vs. non-RA (*p* < 0.001), RA vs. the group where arthritis resolved (*p* < 0.001), and RA vs. other inflammatory rheumatic diseases (*p* < 0.004). Patients who developed RA at the onset of UA were more often diagnosed with bone erosions seen on US as compared with the patients in the non-RA group (*p* = 0.004) and patients whose arthritis resolved (*p* < 0.001) (Table 2).

### 3.2. SNP Analysis Results

As regards SNP variants tested, no statistical significance was found between the patients who developed RA and those who developed other disease outcomes (Table 3).

To analyze the significance of SNPs to sociodemographic, clinical, laboratory, and instrumental variables, patients were divided by genotype into two groups: rs2476601 *AA* + *AG* and *GG*; and rs833070 *GG* + *AG* and *AA*; rs6920220 *AA* + *AG* and *GG*.

The SNP minor allele distribution (homozygous and heterozygous) analysis with sociodemographic, clinical, and laboratory findings revealed that patients whose rs6920220 were present with the minor allele (*AA* or *AG*) had a significantly greater number of tender joints (68/28 TJS) (Table 4). 

## 4. Discussion

The association between RA and various genomic variants, i.e., single nucleotide polymorphisms, has been widely studied and described in scientific publications. By carrying out large-scale genomic studies, researchers are trying to identify SNP variants that are relevant to RA [34]. In the present pilot study, the distribution between investigated SNPs (rs2476601 on the *PYTPN22* gene, rs833070 on the *VEGF* gene, and rs6920220 on the *TNFAIP3* gene) in the cohort of patients with early undifferentiated arthritis was assessed, and SNPs’ minor alleles’ association with sociodemographic, clinical, laboratory, ultrasound data and UA outcomes was tested. 

The meta-analysis demonstrated SNP rs2476601 association with the increased risk of developing RA [19]. This meta-analysis showed that *CT* + *TT* genotypes of the *PTPN22* gene were statistically significantly associated with RA in the European population (OR = 1.683) [16]. In the current pilot study, we did not confirm an association between the risk of UA progressing to RA and *PTPN22* gene polymorphism. Moreover, regarding the risk of developing RA associated with rs2476601 polymorphism, the RA group was compared with the non-RA group, which was highly heterogeneous and consisted of the patients whose arthritis resolved during the study period as well as the patients who were later diagnosed with SpA, PsA, SLE, etc. Published literature indicates that this polymorphism is associated with the risk of SLE [35] as well as other autoimmune diseases [20,21,36] and that *PTPN22* rs2476601 genotypes have been found to be associated with exaggerated immune system response and the inflammatory process [37]. This allows us to suggest that the control group in the study was also prone to an exaggerated immune response and maintenance of inflammatory processes and that the results of the study may have been influenced by the comorbidities of the patients studied, as 16% of diabetic patients are known to have *PTPN22* polymorphism, while only 6% of the healthy population have *PTPN22* polymorphism [38,39]. A meta-analysis revealed the association between rs2476601 and RA markers (RF and anti-CCP) [16]. In the present study, the association between this polymorphism and RF and anti-CCP was not confirmed.

There are several published studies supporting an association between *VEGF* gene rs833070 polymorphism and the risk of developing RA [18,23,29]; our literature review identified two studies that analyzed the association between *VEGF* rs833070 polymorphism and RA. The sample sizes referenced in these two studies were relatively small too: 98 RA subjects and 100 controls. Both studies showed an association between *VEGF* polymorphisms and serum VEGF levels. These studies also confirm that the frequency of the *A* allele was higher in RA patients as compared to controls [23,29]. The association between *AA* genotype and serum VEGF levels was also confirmed, as was an association between disease activity (DAS 28) and joint abnormalities seen on US scan [23]. The study analysed the *VEGF* gene rs833070 polymorphism and its association with risk of developing RA, although no assotiation was confirmed. There was found no link between *VEGF* gene rs833070 polymorphism and RA diagnostic markers, such as greater anti-CCP, RF, ESR, CRP values, grade of synovitis seen on US, or serum VEGF levels. 

The rs6920220 polymorphism in the *TNFAIP3* gene is associated with the risk of many autoimmune diseases, including RA [40,41,42]. This polymorphism may regulate immune cells via the protein it encodes and therefore is associated with inflammatory processes and autoimmune diseases [43,44]. A meta-analysis supported the association between *TNFAIP3* polymorphism and the risk of RA, and a statistically significant association was also found in RA patients who were anti-CCP positive [17]. The present study failed to establish the association between *TNFAIP3* polymorphism and the risk of RA but confirmed a significant association between *AA* + *AG* genotypes and the number of painful joints (68 and 28). These results suggest that a larger cohort of patients could help to clarify the genetic origin of arthralgias. In the present pilot study, we did not confirm the association between the polymorphisms studied and the risk of RA in the population with early undifferentiated arthritis. This is most likely due to the small sample size. In addition, the fact that the control group of RA patients was composed of the individuals who also suffered from other inflammatory arthritis (PsA, SpA, SLE, etc.) might have influenced the results as well. As mentioned above, two of the analyzed polymorphisms (rs2476601, rs6920220) are associated with increased immune reactivity and susceptibility to the inflammatory process [37,43,44]. The fact that patients with the SNP rs6920220 *GG* and *GA* genotypes had a significantly greater number of tender joints at the study baseline could suggest that a more comprehensive prospective study on the patients suffering from arthralgia could provide knowledge of whether individuals with these genotypes will later develop inflammatory arthritis. To better understand the role the analyzed polymorphisms play in the etiopathogenesis of inflammatory diseases and the possible relevance to UA outcomes, a large sample of research subjects would be needed, and it would be very useful to include a control group of healthy subjects. The present study is distinctive in that it examined a population of individuals with early undifferentiated arthritis. The cohort studied was homogeneous, with only Caucasians; the cohort was distributed over a relatively small residential area; and there was a difference with the deletions found in the European population studied in the SNP database (the T allele was found in the European population [45], whereas the *G* recessive allele was identified in the population of this pilot study).

## 5. Conclusions

In conclusion, the present study highlighted various clinical, laboratory, and ultrasound differences among the patients with early undifferentiated arthritis based on disease outcomes at follow-up. While associations with the number of tender and swollen joints, RF, anti-CCP positivity, and ultrasound scores were observed, the tested SNP variants did not show any significant associations with disease outcomes. On the other hand, patients with minor alleles (*AA* or *AG*) of rs6920220 exhibited a significantly greater number of tender joints. Further research on larger cohorts is necessary to provide deeper insight into genetic and clinical factors influencing progression of early undifferentiated arthritis to rheumatoid arthritis and other inflammatory rheumatic diseases.

## Figures and Tables

**Table 1 medicina-59-01824-t001:** Sociodemographic, clinical, and laboratory data from early undifferentiated arthritis patient cohort; comparison of sociodemographic, clinical, and laboratory data between rheumatoid arthritis and other patients outcomes groups.

Variable	All UA Patients (*n* = 92)	RA Group (*n* = 27)	Non-RA Group (*n* = 65)	*p* Value *	Arthritis Resolved (*n* = 21)	*p* Value **	Other Rheumatic Inflammatory Disease (*n* = 44)	*p* Value ***
Sociodemographics								
Age, years	44.25 ± 15.82	47.78 ± 16.36	42.79 ± 15.48	0.178	43.10 ± 15.67	0.266	42.34 ± 15.56	0.220
Females, *n* (%)	61 (66.3)	20 (74.1)	41 (63.1)	0.310	14 (66.7)	0.575	27 (61.4)	0.272
BMI, kg/m^2^	24.14 ± 3.43	23.71 ± 3.40	24.32 ± 3.18	0.318	23.74 ± 3.07	0.771	24.60 ± 3.22	0.225
Daily smokers, *n* (%)	14 (15.2)	4 (14.8)	10 (15.4)	0.945	5 (23.8)	0.428	5 (11.4)	0.671
Education, years	13.40 ± 2.19	13.59 ± 1.95	13.32 ± 2.29	0.751	13.67 ± 2.4	0.865	13.16 ± 2.24	0.592
Presence of rheumatic disease in family, *n* (%)	30 (32.6)	12 (44.4)	18 (27.7)	0.119	7 (33.3)	0.435	11 (25.0)	0.671
Clinical data								
Presence of active infection ^1^, *n* (%)	23 (31.5)	4 (14.8)	25 (38.5)	0.084	11 (52.4)	**0.014**	14 (31.8)	0.257
Presence of comorbidities, *n* (%)	62 (67)	21 (77.8)	41 (63.1)	0.171	13 (61.9)	0.230	28 (63.6)	0.211
Duration of joint pain, months	6.74 ± 4.96	6.26 ± 4.68	6.94 ± 5.09	0.396	6.52 ± 3.53	0.497	7.14 ± 5.1	0.432
Duration of joint swelling, months	5.27 ± 3.22	4.85 ± 3.21	5.45 ± 3.23	0.425	5.76 ± 2.91	0.233	5.92 ± 3.39	0.672
Duration of morning stiffness, minutes	68.04 [0–300]	92.78 [0–300]	57.77 [0–300]	0.117	52.86 [0–120]	0.281	30 [0–300]	0.122
Patient’s joint pain VAS, mm	45.42 ± 16.99	43.89 ± 16.18	46.06 ± 17.39	0.431	45.29 ± 16.16	0.684	46.43 ± 18.12	0.391
Patient’s global VAS, mm	47.98 ± 16.40	49.70 ± 18.04	47.26 ± 15.77	0.646	46.86 ± 17.13	0.446	47.46 ± 15.28	0.840
Physician’s global VAS, mm	45.80 ± 13.89	46.07 ± 13.68	45.69 ± 14.08	0.931	44.67 ± 13.79	0.942	46.18 ± 14.35	0.938
66 SJC	3.17 ± 2.14 [1–14]	4.63 ± 2.80 [2–14]	2.57 ± 1.44 [1–8]	**<0.001**	2.19 ± 0.87	**<0.001**	2.75 ± 1.61 [1–8]	**<0.001**
28 SJC	2.73 ± 1.6 [0–8]	3.81 ± 1.71 [2–8]	2.28 ± 1.24 [0–7]	**<0.001**	2.19 ± 1.21	**<0.001**	2.32 ± 1.41 [0–7]	**<0.001**
68 TJC	7.33 ± 5.77 [0–28]	9.59 ± 6.03 [1–28]	6.39 ± 5.43 [0–28]	**0.010**	4.95 ± 4.23	**0.003**	7.07 ± 5.84 [0–28]	0.060
28 TJC	5.37 ± 3.75 [0–18]	6.70 ± 3.75 [1–14]	4.82 ± 3.63 [0–18]	**0.021**	4.00 ± 3.22	**0.015**	5.20 ± 3.78 [0–18]	0.071
HAQ	0.63 ± 0.41	0.742 ± 0.47	0.58 ± 0.38	0.106	0.51 ± 0.32	0.052	0.61 ± 0.40	0.258
DAS 28	4.50 ± 1.09	5.01 ± 0.86	4.29 ± 1.11	**0.005**	4.67 ± 1.18	0.313	4.11 ± 1.05	**<0.001**
Laboratory findings								
ESR, mm/h	32.55 [2–144]	39.67 ± 34.51	29.45 ± 23.85	0.239	32.89 ±25.75	0.647	27.93 ± 23.11	0.178
CRP, mg/L	17.65 [0.16–144.30]	22.42 [0.39–111.50]	15.67 [0.16–144.30]	0.065	18.99 [0.30–84.90]	0.430	4.6 [0.16–144.30]	0.036
RF positive, *n* (%)	31 (33.7)	24 (88.9) (*n* = 27)	7 (10.8) (*n* = 65)	**<0.001**	2 (9.5) (*n* = 21)	**<0.0005**	5 (11.4) (*n* = 44)	**<0.001**
Anti-CCP ^2^ positive, *n* (%)	30 (36.1)	24 (89.9) (*n* = 27)	6 (10.7) (*n* = 56)	**<0.001**	1 (5.0) (*n* = 19)	**<0.0005**	5 (13.5) (*n* = 37)	**<0.001**
RF and anti-CCP positive, *n* (%)	27 (31.0)	22 (81.5)	5 (8.9)	**<0.001**	1 (5.0)	**<0.0005**	4 (10.8)	**<0.001**
HLA B27 ^3^ expressed, *n* (%)	20 (28.2)	1 (6.3) (*n* = 16)	19 (34.5) (*n* = 55)	**0.010**	3 (17.6) (*n* = 17)	0.321	16 (42.1) (*n* = 38)	**0.010**
ANA ^4^ positive, *n* (%)	19 (45.2)	5 (38.5) (*n* = 13)	14 (48.3) (*n* = 29)	0.238	1 (14.3) (*n* = 7)	0.260	13 (59.1) (*n* = 22)	0.555
VEGF ^5^, pg/mL	516.16 [25.75–3438.25]	697.60 [25.75–3438.25] (*n* = 23)	437.42 [53.97–1854.91] (*n* = 53)	0.086	449.50 [53.97–1437.35] (*n* = 20)	0.242	283.42 [58.47–1854.91] (*n* = 33)	0.083

Continuous data are presented in median [minimum and maximum] values or mean ± standard deviation, counted as numbers and valid percentages. BMI, body mass index; VAS, visual analogue scale; SJC, swollen joint count; TJC, tender joint count; HAQ, Health Assessment Questionnaire; DAS 28, Disease Activity Score 28 using erythrocyte sedimentation rate; ESR, erythrocyte sedimentation rate; CRP, C-reactive protein; RF, rheumatoid factor; anti-CCP, anti-citrullinated protein antibodies; HLA B27, human leukocyte antigen B27; ANA, antinuclear antibodies, VEGF, vascular endothelial growth factor; UA, early undifferentiated arthritis; RA, rheumatoid arthritis; non-RA group, patients who at a 12-month follow-up had not developed rheumatoid arthritis; other rheumatic inflammatory diseases, patients who at a 12-month follow-up were diagnosed with psoriatic arthritis, ankylosing spondylitis, systemic lupus erythematosus, etc. *p* significant if <0.05. Total tested: ^1^—N = 62, ^2^—N = 82, ^3^—N = 72, ^4^—N = 42, ^5^—N = 76. * *p* value between RA and non-RA groups; ** *p* value between RA group and patients whose arthritis resolved; *** *p* value between RA group and patients’ group who developed other inflammatory rheumatic diseases, statistically significant data is bolded.

**Table 2 medicina-59-01824-t002:** Ultrasound findings in early undifferentiated arthritis patients cohort at study baseline; comparison of ultrasound findings between rheumatoid arthritis and other patients outcomes groups.

Variable	All UA Patients (n = 92)	RA Group (n = 27)	Non-RA Group (n = 65)	*p* Value *	Arthritis Resolved (n = 21)	*p* Value **	Other Rheumatic Inflammatory Disease (n = 44)	*p* Value ***
Synovitis score	6.35 ± 4.58	9.78 ± 5.20	4.92 ± 3.44	**<0.001**	3.81 ± 2.21	**<0.001**	5.46 ± 3.81	**<0.001**
Power Doppler score	2.0 [0–14]	6.0 [0–14]	1.0 [0–10]	**<0.001**	1.0 [0–4]	**<0.001**	2.0 [0–10]	**<0.001**
Erosions (grade) median	0.0 [0–10]	2.0 [0–8]	0.0 [0–10]	**<0.001**	0.0 [0,1]	**<0.001**	0.0 [0–10]	**0.004**
Presence of erosions, n (%)	34 (37.0)	16 (59.3)	18 (27.7)	**0.004**	2 (9.5)	**<0.001**	16 (36.4)	0.060

Continuous data are presented in median [minimum and maximum] values or mean ± standard deviation counted as numbers and valid percentages. UA, early undifferentiated arthritis; RA, rheumatoid arthritis; non-RA group, patients who at a 12-month follow-up had not developed rheumatoid arthritis; other rheumatic inflammatory disease, patients who at a 12-month follow-up had been diagnosed with psoriatic arthritis, ankylosing spondylitis, systemic lupus erythematosus, etc. *p* significant if <0.05. * *p* value between RA and non-RA groups; ** *p* value between RA group and patients whose arthritis resolved; *** *p* value between RA group and patients’ group who developed other inflammatory rheumatic diseases, statistically significant data is bolded.

**Table 3 medicina-59-01824-t003:** Association between rs2476601, rs833070, and rs6920220 polymorphisms and risk of developing rheumatoid arthritis in early undifferentiated arthritis cohort.

SNP	Chr	Position Base Pair	Minor Allele	MAF	N	Odds Ratio (OR)	*p* Value
rs2476601	1	113834946	A	0.1	92	0.99	0.98
rs833070	6	43774889	G	0.4	92	1.0	0.97
rs6920220	6	137685367	A	0.1	92	0.48	0.13

SNP, single nucleotide polymorphism; Chr, chromosome; MAF, minor allele frequency; N, number of patients; OR, odds ratio. *p* significant if <0.05.

**Table 4 medicina-59-01824-t004:** Association between rs2476601, rs833070, and rs6920220 polymorphisms and sociodemographic, clinical, laboratory, and ultrasound data in the cohort of patient with early undifferentiated arthritis.

	Single Nucleotide Polymorphisms	rs2476601		rs833070		rs6920220	
Variable			*p* Value		*p* Value		*p* Value
Patients’ distribution based on genotype		*AA* + *AG/GG* 23/69	-	*GG* + *GA/AA* 16/76	-	*AA* + *AG/GG* 30/62	-
66 SJC		2.96 ± 1.88; 2 [1–8] *	0.619	3.00 ± 1.90; 3 [1–8] *	0.485	3.1 ± 2.68; 2 [1–14] *	0.171
		3.27 ± 2.26; 3 [1–14] *		3.25 ± 2.24; 2 [1–14] *		3.21 ± 1.84; 3 [1–12] *	
28 SJC		2.46 ± 1.32; 2 [1–5] *	0.718	2.48 ± 1.34; 2 [1–5] *	0.385	2.53 ± 1.59; 2 [1–8] *	0.262
		2.84 ± 1.72; 2 [0–8] *		2.83 ± 1.71; 2 [0–8] *		2.82 ± 1.62; 2 [0–8] *	
68 TJC		6.61 ± 4.62; 6 [1–17] *	0.368	6.78 ± 4.62; 6 [1–17] *	0.901	5.87 ± 6.25; 4 [1–28] *	**0.019**
		7.64 ± 6.21; 6 [0–28] *		7.55 ± 6.21; 6 [0–28] *		8.03 ± 5.44; 7 [0–24] *	
28 TJC		4.96 ± 3.05; 4 [1–14] *	0.654	5.07 ± 3.05; 5 [1–14] *	0.836	4.37 ± 3.84; 3 [0–18] *	**0.025**
		5.54 ± 4.02; 4 [0–18] *		5.49 ± 4.02; 4 [0–18] *		5.85 ± 3.63; 5 [0–14] *	
ESR, mm/h		29 [2–144]	0.754	27.0 [2–144]	0.590	22.0 [2–110]	0.853
		17 [2–88]		19.50 [2–88]		27.0 [2–144]	
CRP, mg/L		5.03 [0.20–111.50]	0.732	6.47 [0.20–111.50]	0.646	4.3 [0.16–111.50]	0.500
		4.48 [0.16–144.30]		4.35 [0.16–144.30]		7.32 [0.20–144.30]	
RF, kU/L		20.0 [20.00–814.20]	0.637	20.0 [20.00–814.20]	0.417	20 [9.59–814.20]	0.674
		20.0 [9.59–306.00]		20.0 [9.59–306.00]		20 [20.00–240.30]	
RF positive (>30 kU/L)		11 (39.3)	0.702	11 (40.7)	0.357	9 (30.0)	0.602
		20 (31.3)		20 (30.8)		22 (35.5)	
Anti-CCP, U/L		2.0 [2–300]	0.136	2.0 [2–300]	0.387	2.0 [2–300]	0.871
		2.0 [2–300]		2.0 [2–300]		2.0 [2–300]	
Anti-CCP positive (≥5 U/mL)		11 (44.0)	0.134	10 (41.7)	0.384	10 (33.3)	0.870
		19 (30.6)		20 (31.7)		20 (35.1)	
RF and anti-CCP positive		10 (40.0)	0.991	10 (41.7)	0.186	9 (30.0)	0.880
		17 (27.4)		17 (27.0)		18 (31.6)	
HLA B27 expressed		6 (28.6)	0.516	6 (30.0)	0.830	5 (23.8)	0.597
		14 (28.0)		14 (27.5)		15 (30.0)	
ANA positive		7 (46.7)	0.327	6 (42.9)	0.826	4 (26.7)	0.071
		12 (44.4)		13 (46.4)		15 (55.6)	
VEGF, pg/mL		384.64 [25.75–3438.23]	0.563	374.37 [68.67–3438.23]	0.773	284.91 [115.88–3438.23]	0.323
		312.46 [68.67–1854.91]		272.30 [25.75–1854.91]		414.47 [25.75–1967.50]	
Synovitis, scores		6.29 ± 4.22; 5 [2–16] *	0.806	6.37 ± 4.28; 5 [2–16] *	0.938	5.87 ± 5.31; 4 [2–25] *	0.215
		6.37 ± 4.77; 5 [2–25] *		6.34 ± 4.74; 4 [2–25] *		6.5 ± 4.22; 5 [2–21] *	
PD, scores		2.79 ± 2.84; 2 [0–10] *	0.627	2.89 ± 2.85; 2 [0–10] *	0.798	2.70 ± 3.26; 2 [0–12] *	0.164
		3.17 ± 3.13; 3 [0–14] *		3.12 ± 3.14; 2 [0–14] *		3.23 ± 2.94; 3 [0–14] *	
Erosions, scores		1.18 ± 1.89; 0 [0–6] *	0.722	1.22 ± 1.91; 0 [0–6] *	0.988	1.03 ± 1.90; 0 [0–8] *	0.468
		1.31 ± 2.23; 0 [0–10] *		1.29 ± 2.22; 0 [0–10] *		1.39 ± 2.23; 0 [0–10] *	
Erosions detected		9 (32.1)	0.803	9 (33.3)	0.643	9 (30.0)	0.336
		25 (39.1)		25 (38.5)		25 (40.3)	

Continuous data are presented in median [minimum and maximum] values or mean ±standard deviation counted as numbers and valid percentages. SJC, swollen joint count; TJC, tender joint count; ESR, erythrocyte sedimentation rate; CRP, C-reactive protein; RF, rheumatoid factor; anti-CCP, anti-citrullinated protein antibodies; HLA B27, human leukocyte antigen B27; ANA, antinuclear antibodies; VEGF, vascular endothelial growth factor; PD, Power Doppler * data is presented in both mean and median values., statistically significant data is bolded. *p* significant if <0.05.

## Data Availability

The original datasets are not publicly available due to data protection policies. The data presented in this study are available on scientific request from the corresponding author.

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
