# Peer review of "Association between PYTPN22 rs2476601, VEGF rs833070, TNFAIP3 rs6920220 Polymorphisms and Risk for Rheumatoid Arthritis in Early Undifferentiated Arthritis Patients: A Pilot Study"

_medicina, 2023, doi:10.3390/medicina59101824_

Round 1
Reviewer 1 Report
The study evaluates 3 single-nucleotide polymorphisms in different genes, identifying one of them associated with disease activity (Undifferentiated Arthritis) in a cohort of patients who were assessed for 12 months to classify patients based on the outcome at the end of the follow-up period.
Certainly, the article is suitable for the journal. I have a few comments to help enhance and better understand the authors' message, as listed below:
1.- Please verify the order of the tables, as Table 2 is mentioned before Table 1.
2.- Verify grammar and writing in the text; there are multiple instances of "))."
3.- Please clarify the rationale for comparing only with the RA group. I suggest considering the inclusion of a group without joint disease to reliably demonstrate the genotype and allele frequencies in your population, or adding a group of patients with previously diagnosed RA, as the patients in the compared group might be among those who exhibited "Arthritis resolved."
4.- In the statistical section, the potential for bias in making multiple comparisons between only two groups should be addressed. You may consider employing a statistical method that allows for comparison among multiple groups and identifies statistically different groups, reducing errors in comparisons.
5.- Finally, quantitative variables could be evaluated using different methods within each group. While significant changes are observed between the 3 groups in Table 2, they are not identified in Table 4 (this may be an effect of the stratification performed in Table 2, which is not respected in Table 4).
6.- I suggest conducting a statistical power calculation to assess the strength of the association identified in your study cohort.
Check the grammar overall.
Author Response
Dear reviewer, thank You for time You have spent reading this article and Your comments on it. It was very valuable and helped us to improve our manuscript. Below we are providing comments regarding Your remarks and questions.
1.- Please verify the order of the tables, as Table 2 is mentioned before Table 1.
Table numeration has been corrected.
2.- Verify grammar and writing in the text; there are multiple instances of "))."
Grammar in the text was corrected and other typos were corrected too.
3.- Please clarify the rationale for comparing only with the RA group. I suggest considering the inclusion of a group without joint disease to reliably demonstrate the genotype and allele frequencies in your population, or adding a group of patients with previously diagnosed RA, as the patients in the compared group might be among those who exhibited "Arthritis resolved."
Thank you for Your suggestion about considering the inclusion of a group without joint disease. We have commented on this limitation of our pilot early undifferentiated arthritis study in the discussion section. We decided to compare only with rheumatoid arthritis (RA) group based on our study sample size. Patients who were included into “arthritis resolved” group had a very low chance to be diagnosed with RA. Patients whose disease outcome was confirmed “arthritis resolved” during 12 months follow up period had no signs of arthritis at least for 6 months and without need of steroids and DMARDs. This patients group from the beginning of this study did not meet classification criteria for RA, and arthritis resolved and during all follow up period had at least 6 months of drug free remission. This is why is why it is a very low chance for these patients to develop RA in the future.
4.- In the statistical section, the potential for bias in making multiple comparisons between only two groups should be addressed. You may consider employing a statistical method that allows for comparison among multiple groups and identifies statistically different groups, reducing errors in comparisons.
Thank You for Your insights on statistical analysis. In this study our target group were patients who developed rheumatoid arthritis. Due to this we applied statistical methods that were listed in statistics section. These methods helped us to confirm the differences between RA patients group and other early undifferentiated arthritis outcome groups separately (data presented in table 1 and table 2). In table 1 we present statistically significant results that are key features of RA (number of tender and swollen joints, RF and anti-CCP positivity), and in table 2 - active inflammation present in RA patients synovium. And as the aim of this study was to determine which tested SNPs are associated with risk to develop RA, this is why we did not analyses statistical differences between all outcome groups all together.
5.- Finally, quantitative variables could be evaluated using different methods within each group. While significant changes are observed between the 3 groups in Table 2, they are not identified in Table 4 (this may be an effect of the stratification performed in Table 2, which is not respected in Table 4).
In table 2 we presented data based on the same statistical methods as it was applied in table 1. Data presented separately as it shows results of instrumental test. We did not performed the same stratification in table 4, as the aim of this table is to show the association between tested SNPs alleles and symptoms, laboratory, and ultrasound findings that are assessed in early undifferentiated arthritis diagnostics.
If we had performed stratification between three UA outcome groups and assessed association with tested SNPs alleles, the sample sizes of the groups would have been too small to obtain reliable statistics.
6.- I suggest conducting a statistical power calculation to assess the strength of the association identified in your study cohort.
Larger sample size of patients based on power calculation results and this pilot study data has to be tested in the future.

Reviewer 2 Report
This study investigates the association between rs2476601, rs833070, and rs6920220 SNPs and risk of Rheumatoid arthritis progression in undifferentiated arthritis patients.
The following points need to be addressed carefully:
1. The title of this manuscript can be modified to “Association of PYTPN22 rs2476601, VEGF rs833070, TNFAIP3 rs6920220 Polymorphisms and Risk of Rheumatoid Arthritis in Early Undifferentiated Arthritis Patients: A Pilot Study”
2. The authors should elaborate the rationale behind choosing these specific 3 SNPs for this study and provide the details in the introduction section.
3. The authors should also mention the hypothesis of this study before stating the aim of this study in the introduction section.
4. The authors gave no detailed description of the TNFAIP3 rs6920220 in the introduction section (Line 78-79).
5. In Line 82, there is repetition of the word laboratory, please correct it.
6. The authors are requested to describe 68TJC/66SJC joint count definition somewhere in the manuscript.
7. The authors should mention regarding the company of the ELISA kits and other methodologies used in the manuscript.
8. In Line 374, kindly mention the citation and relevant reference.
9. The discussion section requires a thorough revision to support the results of this study, along with that kindly mention the limitations of the study.
10. This study investigates the association between rs2476601, rs833070, and rs6920220 SNPs and risk of Rheumatoid arthritis progression in undifferentiated arthritis patients.
The following points need to be addressed carefully:
1. The title of this manuscript can be modified to “Association of PYTPN22 rs2476601, VEGF rs833070, TNFAIP3 rs6920220 Polymorphisms and Risk of Rheumatoid Arthritis in Early Undifferentiated Arthritis Patients: A Pilot Study”
2. The authors should elaborate the rationale behind choosing these specific 3 SNPs for this study and provide the details in the introduction section.
3. The authors should also mention the hypothesis of this study before stating the aim of this study in the introduction section.
4. The authors gave no detailed description of the TNFAIP3 rs6920220 in the introduction section (Line 78-79).
5. In Line 82, there is repetition of the word laboratory, please correct it.
6. The authors are requested to describe 68TJC/66SJC joint count definition somewhere in the manuscript.
7. The authors should mention regarding the company of the ELISA kits and other methodologies used in the manuscript.
8. In Line 374, kindly mention the citation and relevant reference.
9. The discussion section requires a thorough revision to support the results of this study, along with that kindly mention the limitations of the study.
10. The manuscript requires editing in order to improve the grammatical and language errors.
The manuscript requires editing in order to improve the grammatical and language errors.
Author Response
Dear reviewer, thank Your for Your time that You have spent reading our manuscript. Your ideas and commers were very valuable and helped us to improve our manuscript. Below we provide our comments and answers.
- The title of this manuscript can be modified to “Association of PYTPN22 rs2476601, VEGF rs833070, TNFAIP3 rs6920220 Polymorphisms and Risk of Rheumatoid Arthritis in Early Undifferentiated Arthritis Patients: A Pilot Study”.
Thank You for Your suggestion. We decided to modify the title of the manuscript based on Your recommendation.
- The authors should elaborate the rationale behind choosing these specific 3 SNPs for this study and provide the details in the introduction section.
Information provided: lines 65 – 66.
- The authors should also mention the hypothesis of this study before stating the aim of this study in the introduction section.
The hypothesis has been added: line 84-85.
- The authors gave no detailed description of the TNFAIP3 rs6920220 in the introduction section (Line 78-79).
More detailed description has been added: lines 82-83.
- In Line 82, there is repetition of the word laboratory, please correct it.
Corrected.
- The authors are requested to describe 68TJC/66SJC joint count definition somewhere in the manuscript.
SJC66/TJC68 is a tool for the measurement of the peripheral arthritis component of the MSK disease activity domain. Reference has been added in the manuscript line 125.
- The authors should mention regarding the company of the ELISA kits and other methodologies used in the manuscript.
We have added company names of ELISA kits that were used. Lines 139, 141-142. Other methodologies used in the manuscript were mentioned before.
- In Line 374, kindly mention the citation and relevant reference.
Link to https://www.ncbi.nlm.nih.gov/snp/ website has been added.
- The discussion section requires a thorough revision to support the results of this study, along with that kindly mention the limitations of the study.
Discussion has been revised and more information added.
- The manuscript requires editing in order to improve the grammatical and language errors.
Gramma has been improved and text edited.

Reviewer 3 Report
1. Introduction
2. Patients and Methods
2.1. Patients
A flow chart should be created
3. Results
3.1. Patient characteristics
Should table 2 be table 1 ?
Table 2; ankylosing spondylitis instead of ankylosing spondylarthritis
Were they divided into two or three groups? (patients were divided by genotype into two groups: rs2476601 AA+AG and GG; rs833070 GG+AG and AA; rs6920220 AA+AG and GG). (Table 4).
Similarity index 20% (should be reduced).
Author Response
Dear reviewer, thank Your for Your time that You have spent reading our manuscript. Your ideas and commers were very valuable and helped us to improve our manuscript. Below we provide our comments and answers.
- Should table 2 be table 1 ?
Number of tables has been corrected.
- Table 2; ankylosing spondylitis instead of ankylosing spondylarthritis
Corrected
- Were they divided into two or three groups? (patients were divided by genotype into two groups: rs2476601 AA+AG and GG; rs833070 GG+AG and AA; rs6920220 AA+AG and GG). (Table 4).
Based on tested SNPs alleles patients were divided into two groups for each SNP:
- rs2476601 AA+AG and GG
- rs833070 GG+AG and AA
- rs6920220 AA+AG and GG
- Similarity index 20% (should be reduced).
We applied a plagiarism-checking tool to detect potentially plagiarized content. A few parts were corrected and marked in the text. However, several internationally acclaimed terms such as body mass index, C-reactive protein, erythrocyte sedimentation rate, inflammatory rheumatic disease, analysis of variance, etc. are repeatedly used in our manuscript. We also mention a few titles of organizations, e.g. Vilnius Regional Biomedical Research Ethics Committee. In addition, we detail and cite a questionnaire and some classification criteria. These parts appear as plagiarized but cannot be changed.

Round 2
Reviewer 1 Report
MDPI Medicine
Of the 6 comments made above, I will only address 3 of them, the rest have been addressed or resolved.
Point 3: It is important to mention this answer in the manuscript because it is part of the justification of your study, although it is an answer, there is still a gap in the manuscript.
Point number 5: I will detail my observation to each variable that I feel could be improved, within the analysis of the quantitative variables that you mention in the methods that you employed Kruskal-Wallis, in what analysis did you perform this?, part of the observation made in the previous round is focused on the multiple subgroup comparisons made.
Some of the variables, for example in table 2 in the degree of erosion, there are values of 0 with a range of 0-10 (all UA patients) and 2 [0-8] RA group, is the comparison of these ranges underestimating the significance value?
In the results by genotype, I understand that the genetic analysis model used was selected according to the MAF, but the rs833070, although its frequency is close to 0.4, was the analysis done correctly, since the sample sizes would indicate that it takes A as a minor allele, and this modifies the hypothesis that was raised with the other 2 SNV.
Finally, point number 6: Despite being a pilot study you identify several interesting findings for clinical practice (or possible use of the genetic markers in conjunction with the broad clinical they evaluated), the main genetic finding mentioned in the conclusion is rs6920220, with this finding you can evaluate the statistical power you have from your pilot sample to indicate the strength of the results obtained so far, although in the limitations you already mention the sample size, it is important to mention that the strength of association of your study allows you to reach your conclusion.
Or describe the purpose of the study, i.e. whether you are disclosing the collection and identification methods, sampling method, etc. for exploratory purposes only, without reaching the conclusion you mention.
No additional comments
Author Response
Dear reviewer, thank You for Your time and comments to our manuscript. We are very grateful that Your provided us all comments and ways to improve it. Here below we provide answers to Your questions.
Point 3: It is important to mention this answer in the manuscript because it is part of the justification of your study, although it is an answer, there is still a gap in the manuscript.
Information about patients whose arthritis resolved was provided in lines 110-112, and more information has been added into manuscript (lines 112-113).
Point number 5: I will detail my observation to each variable that I feel could be improved, within the analysis of the quantitative variables that you mention in the methods that you employed Kruskal-Wallis, in what analysis did you perform this?, part of the observation made in the previous round is focused on the multiple subgroup comparisons made.
We apologize for leaving this test in statistical analysis section. We did not add results in this manuscript that were obtained applying Kruskal-Wallis test.
Some of the variables, for example in table 2 in the degree of erosion, there are values of 0 with a range of 0-10 (all UA patients) and 2 [0-8] RA group, is the comparison of these ranges underestimating the significance value?
Even though the ranges in ultrasound analysis in some cases were [0-10] in UA patients case, or [0-8] in RA patients case, next to these ranges we provide median. In statistical analysis median values in each group were calculated. We applied tests for non – parametric data distribution that do not assume a specific distribution and can be used when comparing groups with different ranges. In statistical analysis of ultrasound data from RA and non-RA group, RA and patients whose arthritis resolved completely and RA and patients who developed other rheumatic inflammatory disease groups were compared. And only in RA group at the disease onset median of erosions grade was 2, in other mentioned groups it was 0. In RA group patients at the disease onset significantly more often were present with erosions. And erosions detection at the disease onset is poor prognosis sign for RA. The same results tendency is observed in Power Doppler results. These results once again proves that RA group really consists of typical RA patients.
In the results by genotype, I understand that the genetic analysis model used was selected according to the MAF, but the rs833070, although its frequency is close to 0.4, was the analysis done correctly, since the sample sizes would indicate that it takes A as a minor allele, and this modifies the hypothesis that was raised with the other 2 SNV.
We thank the reviewer for bringing out this point. The analysis is correct, and G is the minor allele. However, there was a typo in the numbers for the genotypes. The correct sample sizes for genotypes GG and GA/AA are 16 and 76, respectively. This has now been corrected in Table 4.
Finally, point number 6: Despite being a pilot study you identify several interesting findings for clinical practice (or possible use of the genetic markers in conjunction with the broad clinical they evaluated), the main genetic finding mentioned in the conclusion is rs6920220, with this finding you can evaluate the statistical power you have from your pilot sample to indicate the strength of the results obtained so far, although in the limitations you already mention the sample size, it is important to mention that the strength of association of your study allows you to reach your conclusion.
Or describe the purpose of the study, i.e. whether you are disclosing the collection and identification methods, sampling method, etc. for exploratory purposes only, without reaching the conclusion you mention.
Thank You for Your observations and comments regarding our study results. Based on power size calculator comparing two groups two tailed, where effect size is 0.5, power (1-β err prob) 0.8, sample size is 102. Our sample size is nearly as recommended based on power calculation. Based on statistical power calculations we can conclude that rs6920220 is significantly associated with the number of tender joints. Although a larger sample is needed to confirm tested SNPs association with a risk to UA progress into RA.
